# Ivermectin Augments the Anti-Cancer Activity of Pitavastatin in Ovarian Cancer Cells

**DOI:** 10.3390/diseases11010049

**Published:** 2023-03-14

**Authors:** Mohammed Jasim Jawad, Alan Richardson

**Affiliations:** 1The School of Pharmacy and Bioengineering, Guy Hilton Research Centre, Keele University, Thornburrow Drive, Stoke-on-Trent ST4 7QB, Staffordshire, UK; mohammed.j@uokerbala.edu.iq; 2College of Veterinary Medicine, Kerbala University, Karbala 56001, Iraq

**Keywords:** pitavastatin, ovarian cancer, ivermectin

## Abstract

We have previously shown that pitavastatin has the potential to be used to treat ovarian cancer, although relatively high doses are likely to be necessary. One solution to this problem is to identify drugs that are synergistic with pitavastatin, thereby reducing the dose that is necessary to have a therapeutic effect. Here, we tested combinations of pitavastatin with the anti-parasitic drug ivermectin in six ovarian cancer cell lines. When tested on its own, ivermectin inhibited the growth of the cells but only with modest potency (IC_50_ = 10–20 µM). When the drugs were combined and assessed in cell growth assays, ivermectin showed synergy with pitavastatin in 3 cell lines and this was most evident in COV-318 cells (combination index ~ 0.6). Ivermectin potentiated the reduction in COV-318 cell viability caused by pitavastatin by 20–25% as well as potentiating apoptosis induced by pitavastatin, assessed by activation of caspase-3/7 (2–4 fold) and annexin-labelling (3–5 fold). These data suggest that ivermectin may be useful in the treatment of ovarian cancer when combined with pitavastatin, but methods to achieve an adequate ivermectin concentration in tumour tissue will be necessary.

## 1. Introduction

Our understanding of the biological basis of cancer has increased enormously over the past decades. The sequencing of the human genome has led to the development of technologies that have been used to identify the molecular drivers of cancers. In turn, this has led to the rational design of molecularly targeted therapies that inhibit these drivers. However, genetic instability in cancer provides a source of variability that allows tumour evolution and the evasion of these therapies. The resulting clinical resistance to targeted therapies means additional therapeutic strategies are still required.

Ovarian cancer is an example of a cancer that is inadequately treated. It affects approximately 1 in 70 women and the prognosis for those diagnosed with advanced disease is disappointing, only around 40% of these patients survive for 5 years or more after diagnosis [1]. Although patients with early-stage disease are usually treated effectively, most patients present with late-stage disease, due to the lack of clear symptoms in the early stages and the lack of an appropriate screening programme [2]. For these patients with advanced disease, surgery and adjuvant chemotherapy, comprising carboplatin and paclitaxel, are the mainstay of therapy. Unfortunately, the disease often recurs. Although these recurrent tumours may respond again to chemotherapy, ultimately resistance to the drugs develops. At this stage, alternative therapies are needed. Recent years have seen the development of new therapeutics and PARP inhibitors have attracted particular attention. These drugs are being introduced into standard care and through synthetic lethality show particular activity in patients with defects in the homologous recombination DNA repair pathway [3]. Excitingly, data from the SOLO clinical trial are sufficiently mature to provide data for patients who have received olaparib for a prolonged period. These data show that 67% of patients who received olaparib as a maintenance therapy were alive after 7 years, whereas 47% of patients who received placebo were alive; median overall survival data are not yet available [2]. Survival data from other PARP inhibitors are eagerly awaited. Despite this exciting success, additional therapies are still needed to treat those patients who do not respond to PARP inhibitors, particularly those with a functional homologous recombination pathway.

One alternative to developing new drugs to treat cancer is to repurpose existing drugs that are currently used to treat other diseases. This is an attractive approach because the time and financial cost of developing the drug is drastically reduced. Furthermore, if the drug has been in clinical use for some time, there is likely to be already a good understanding of the issues associated with the drug. However, many researchers have turned to drug repurposing as a strategy without addressing many of the concerns that form an inherent part of the process of the discovery of a new drug. It cannot be assumed that a drug that is to be repurposed can be used in the same fashion as in its original indication [4]. Pharmacodynamic and pharmacokinetic studies are needed to understand the drug concentration and exposure profile necessary for efficacy in the new disease setting, as it may differ substantially from that used in the original one.

We, and many others, have explored the potential for statins to be repurposed from the treatment of hypercholesterolaemia for use as anti-cancer agents [4,5]. Numerous laboratory studies have shown statins exert cytotoxic effects against cancer cells. The cytotoxicity of statins towards cancer cells reflects their ability to block the synthesis of mevalonate, which serves as a precursor for the isoprenoids farnesol and geranylgeraniol [6]. These isoprenoids are necessary for the post-translational modification of ras family small GTPases, many of which are oncogenes. Thus, the activity of statins as anticancer agents is likely to reflect the inhibition of the processes controlled by these GTPases. However, a more detailed mechanistic basis for statins’ activity remains elusive. Many studies have shown that geranylgeraniol, but not farnesol, can suppress the cytotoxic activity of statins, suggesting that the key proteins affected by statins are substrates of geranylgeranyl transferase. However, the precise identity of the proteins that must be affected by statins for the drugs to exert their cytotoxic activity remains unknown.

We have shown that one statin, pitavastatin, can inhibit the growth of ovarian cancer xenografts, but only if the diet is modified to eliminate geranylgeraniol which otherwise bypasses the blockade of hydroxymethyl glutaryl coenzyme A reductase (HMGCR) that catalyses the synthesis of mevalonate [6,7,8]. Despite robust preclinical data showing that statins have anti-cancer activity, statins have yet to show significant activity in clinical trials. We have argued that this reflects the poor design of clinical trials which have uniformly failed to address all the factors we predict influence the clinical activity of statins [4]. This failure includes poor choice of statin, inappropriate dose, inappropriate dosing interval, and failure to consider dietary sources of geranylgeraniol. Lipophilic statins are typically more potent as anti-cancer agents than hydrophilic ones, most probably due to their improved membrane permeability [6,9]. A long half-life statin, coupled with a suitable dosing interval to maintain suppression of HMGCR, is also likely to be necessary, otherwise mevalonate resynthesis can restart between drug doses [6]. A high dose is also likely to be necessary, significantly above those used to treat hypercholesterolaemia, to obtain the concentrations that are cytotoxic to cancer cells [6]. We consider pitavastatin the statin most likely to be effective as an anti-cancer agent by virtue of its lipophilicity, potency, and relatively long half-life. However, relatively high doses of pitavastatin are likely to be required [7].

The major adverse event associated with statins is myopathy. While this is relatively infrequent when statins are used to treat hypercholesterolaemia, at the elevated doses that are likely to be necessary for cancer therapy, the risk of myopathy is likely to be increased. Thus, one concern with the use of statins as anticancer agents is whether there will be an adequate therapeutic window. While we await improved clinical trials, we have sought to identify additional drugs which could augment the cytotoxicity of statins with the goal that these drugs might increase the therapeutic window for pitavastatin. We have shown that inhibitors of PI 3-kinase [10], Bcl2-family proteins [10], farnesyl diphosphate synthase [11], and prednisolone [12] are all able to augment the activity of statins in preclinical models. Although synergy between statins and other agents has been shown by many other researchers [13,14], it remains desirable to identify additional drugs which potentiate the activity of statins. Repurposing drugs that are already in clinical use is one way to achieve that.

Ivermectin is a member of the avermectin family of lactone macrolides [15,16]. These drugs are used as pesticides, insecticides, and anti-parasitic agents in animals in which they inhibit glutamate-activated chloride channels. Recent studies have suggested that ivermectin, which is already used in humans for the treatment of rosacea, threadworm, scabies and river blindness, could be redeployed to the treatment of cancer [17]. Ivermectin inhibits many pathways that act as drivers of cancer including Wnt [18], YAP [19], and the PAK/Akt/mTORC1 axis controlling autophagy [20,21]. Ivermectin also inhibits importinβ, resulting in cell-cycle arrest and apoptosis [22,23]. Importinβ regulates the levels of several cell-cycle regulators such as p21, p27, and APC/C. Repression of the expression of *KPNB1*, the gene-encoding importinβ, decreases the effectiveness of ivermectin and thereby identifies importinβ as a key target of the drug. Ivermectin also induces ROS and genes involved in maintaining stem cell features [24,25]. Lastly, ivermectin inhibits PgP, allowing it to synergize with cytotoxic chemotherapy [26] by inhibiting cellular export of the chemotherapeutic agents. Considering that we have previously shown statins also affect several of these processes, we hypothesized that ivermectin might be synergistic with paclitaxel. Here, we tested that hypothesis. To the best of our knowledge, the combination of pitavastatin and ivermectin alone has not previously been reported. We show that ivermectin has mostly an additive or synergistic interaction with pitavastatin in a panel of six ovarian cancer cell lines. Synergy was most evident in Cov-318 cells and was observed in several assay settings.

## 2. Methods

Ovcar4, Ovsaho, and Ovcar-8 cells were grown in Roswell Park Memorial Institute (RPMI 1640; Lonza) medium supplemented with 10% fetal bovine serum (FBS; Lonza), 2 mM glutamine (Lonza), and 50 μg/mL penicillin/streptomycin (Lonza). Fuov-1, Cov-318, and Cov-362 were grown in DMEM medium supplemented with 10% fetal bovine serum (FBS; Lonza), 2 mM glutamine (Lonza), and 50 μg/mL penicillin/streptomycin (Lonza). Cells were grown at 37 °C in 5% CO_2_.

For drug combinations studies, cells were collected by trypsinization in 0.05% trypsin/1 mM EDTA, quenched with media, and collected by centrifugation (150× *g*, 3 min). The pellet was resuspended in the culture medium at a concentration of 62,500 cells/mL, with the exception of OVCAR-8 cells which were suspended at a concentration of 25,000 cells/mL. An volume of 80 μL of the suspension was seeded into each well of a 96-well plate and incubated for 24 h at 37 °C in 5% CO_2_. The following day, the cells were treated with pitavastatin and/or 20 μL ivermectin. After a further 72 h, the medium was removed and the cells were fixed with 100 µL 10% TCA, incubated on ice for 30 min, and then washed with water. The cells were stained with 100 μL 0.4% sulforhodamine B (SRB) for 30 min and then washed with 3 × 100 μL 1% acetic acid [27]. After drying, the dye was dissolved in 100 μL 10 mM Tris and A_570_ determined.

Data were analysed by fitting a 4-parameter Hill equation using GraphPad Prism and combination indices (C.I.) determined as:$$C.I.=\frac{{\left[p\right]}_{c}}{{\left[p\right]}_{50}}+\frac{{\left[i\right]}_{c}}{{\left[i\right]}_{50}}$$
where [*p*]*_c_* and [*i*]*_c_* represent the concentrations of pitavastatin and ivermectin in the combination which inhibited growth by 50%, and [*p*]_50_ and [*i*]_50_ represent the concentrations of pitavastatin and ivermectin, respectively, which inhibited growth by 50% when used on their own.

To estimate cell viability, 100,000 cells/well were seeded in 1 mL of growth media in 12-well plates and incubated overnight; subsequently, the cells were exposed to the drugs for 72 h. The supernatant was removed from the cells and retained while the adherent cells were collected by trypsinization as described above. The detached cells were then added to their corresponding supernatants, collected by centrifugation at 150× *g* for 3 min, and the pellet was re-suspended in 0.5 mL of medium. The cells were mixed with an equal volume of 0.4% (*v*/*v*) trypan blue (Sigma-Aldrich, St. Louis, MO, USA) and viable and non-viable cells counted by light microscopy by using a Neubauer haemocytometer. The additive effect of drug combinations was determined using the Bliss independence criterion [28]:E(p,i)=E(p)+E(i)−E(p)×E(i)
where *E (p,i)* is the expected effect of the combination and *E(p)* and *E(i)* are the effect of pitavastatin and ivermectin when used on their own.

To measure caspase activity, cells were collected by trypsinization as described above and resuspended (62,500 cells/mL), and 80 μL was seeded per well in 96-well plates. The cells were incubated at 37 °C overnight. The next day, 20 μL of pitavastatin and/or 20 μL of ivermectin was added to the indicated concentration. The cells were incubated for 72 h at 37 °C. To test caspase 3/7 activity assay, the caspase-3/7 Glo assay (Promega; Madison, WI, USA) was used according to the manufacturer’s instructions and luminescence measured. Considering that the drug treatment might affect the number of cells left after the incubation with the drug, a parallel set of samples was prepared and stained with SRB as described. The caspase-3/7 activity was normalized to the SRB staining to control for any effect of the drugs on cell number.

Alternatively, cells were exposed to drug as described above and apoptosis was assessed by staining with annexin V/propidium iodide using a commercial kit (Miltenyi Biotech). Cells were plated and exposed to drugs as described above. Cells were stained with annexin V/propidium iodide using a commercial kit (Miltenyi Biotech). Cells were washed with 1 mL of 1× annexin V binding buffer and the cells were collected by centrifugation (300× *g*, 10 min). The supernatant was aspirated and the cells were re-suspended in 100 μL of 1× binding buffer. An volume of 10 μL of annexin V was added and the suspension gently mixed and stored in the dark at room temperature for 15 min. After incubation, the cells were washed with 1 mL of 1 × annexin V binding buffer again and collected by centrifugation (300× *g*, 10 min). The supernatant was removed and the cells re-suspended in 500 μL of 1× of binding buffer. To detect dead cells, 5 μL PI solution was added immediately before flow cytometry analysis. Early apoptotic cells were defined as annexin V positive and propidium iodide negative, whereas late apoptotic cells were defined as those stained by both markers.

## 3. Results

To test the hypothesis that ivermectin is synergistic with pitavastatin, we conducted cell growth assays using a panel of ovarian cancer cells each of which were shown to be representative of high-grade ovarian cancer [29]. To select appropriate concentrations of ivermectin for the drug combination studies, we first evaluated the potency of ivermectin on its own. When tested as a single agent, ivermectin inhibited the growth of the cells with comparable potencies, exhibiting IC_50_s between 10 and 20 µM (Table 1). We next performed drug combination studies in which the combination of pitavastatin was varied and the concentration of ivermectin was fixed. Three different fixed concentrations of ivermectin were tested for each cell line and these concentrations on their own inhibited cell growth by 5%, 10%, or 20% (fraction affected, f_a_ = 0.05, 0.1 and 0.2), respectively. In many of the experimental conditions, an additive effect was observed (combination index = 1.0). However, in Fuov-1, COV-362, and COV-318 cells, a synergistic interaction was observed (C.I. < 1.0, Figure 1), although this varied with the concentration of ivermectin. Considering that the greatest synergy was observed in the COV-318 cells, further experiments were performed in this cell line.

To confirm the synergy, we measured the effect of the drug combination on COV-318 cell viability. We initially used trypan blue staining for these studies because this method makes no assumption about the mechanism of cell death. In these studies, the three different concentrations of ivermectin were tested alone and in combination with pitavastatin. The effect of the observed combination was compared with the effect of the combination that would be expected if the drugs had an additive interaction, which was estimated from the Bliss independence criterion. When pitavastatin was combined with each of the three concentrations of ivermectin, an effect was observed that was significantly greater than that expected from an additive interaction, consistent with the synergy observed in cell growth assays (Figure 2).

We have previously shown that pitavastatin induces apoptosis in ovarian cancer cells, raising the possibility that ivermectin may potentiate this. Consistent with our previous observations, pitavastatin activated caspase-3/7 in COV-318 cells. When two separate concentrations were combined, each with two separate concentrations of pitavastatin, more caspase-3/7 activity was observed than in cells exposed to pitavastatin alone (Figure 3). Considering that on its own, ivermectin had no detectable effect on caspase-3/7 activity, no mathematical analysis is required to estimate the expected additive effect of the combination and so these data demonstrate that ivermectin synergizes with pitavastatin to induce apoptosis.

To confirm this, we used flow cytometry to measure the effect of the combination on annexin V labelling as an alternative assessment of apoptosis. Pitavastatin on its own induced apoptosis as shown by a reduction in the number of cells alive as well as a corresponding increase in the number of early apoptotic and late apoptotic cells (Figure 4). On its own, ivermectin had a minimal effect. However, when pitavastatin was combined with ivermectin, there was a substantial increase in the number of early-apoptotic and late-apoptotic cells, confirming the synergistic induction of apoptosis observed in the caspase-3/7 activity.

## 4. Discussion

Here, we explored the potential for ivermectin to interact synergistically with pitavastatin and potentially reduce the dose of pitavastatin necessary to treat ovarian cancer. Previous work from Ricardo et al. has shown a synergistic interaction between pitavastatin, paclitaxel, and ivermectin [30]. These authors found that these drugs synergistically decreased the viability of paclitaxel-resistant cells, using several different models to evaluate synergy. We are particularly interested in whether statins can be used as an alternative to chemotherapy, possibly in patients who have become resistant to chemotherapy, so we explored the combination of pitavastatin and ivermectin in the absence of paclitaxel. In this study, we report synergy between pitavastatin and ivermectin, and this synergy may contribute to the synergy observed in the triple-drug combination evaluated by Ricardo.

We first tested ivermectin as a single agent and confirmed previous observations [22,31] that ivermectin was able to inhibit the growth of ovarian cancer cells with a micromolar potency that was similar between all six cell lines. However, the IC_50_ of ivermectin in these assays was substantially higher than the 50 nM plasma concentration [32] achieved in patients with a normal dose (typically 200 μg/kg) of ivermectin and also higher than the 300 nM achieved in patients treated with a high dose (1500–2000 μg/kg) of ivermectin [33]. Thus, it is doubtful whether ivermectin will show substantial activity against ovarian cancer in patients when used on its own. This is a recurrent theme in drug repurposing. Investigators may report that a drug is a suitable repurposing candidate without adequate consideration of whether it is possible to achieve the necessary concentration of the drug in plasma. A dose that is effective in the treatment of one disease may not be effective in the treatment of a different disease. This is because of either differences in the relationship between pharmacodynamic effect at the drug’s cognate target and efficacy, or because the drug is repurposed to impact an entirely new target at which it has a very different pharmacodynamic effect. For example, ivermectin has been proposed as a treatment for SARS-CoV-2 infection, but when tested in clinical trials, the plasma concentration achieved in patients treated even with a relatively high dose of drug were inadequate to reduce viral load [34].

In the cell growth assays reported here, both additive and synergistic interactions were observed. Although a synergistic effect is desirable, an additive effect of two drugs could be clinically useful if the drugs do not have overlapping toxicity profiles. A synergistic interaction was observed in Fuov-1, Cov-362 and Cov-318 cells. In the latter two cells, synergy was observed at multiple concentrations of ivermectin. The synergy was most pronounced in Cov-318 cells so these were selected for further studies. The drug combination synergistically reduced cell viability and increased apoptosis, as shown by its effect in caspase-3/7 and annexin-V labelling studies. Thus, the evidence for synergy in these cells is robust because it was observed in three different assay formats. However, we only observed synergy in a subset of the cell lines and marked synergy in one cell line. If this combination is to be clinically useful, it would be useful to identify biomarkers which predict the response to the drug combination. This could be facilitated by uncovering of the mechanism underlying the synergy. Although we currently do not understand this mechanism in detail, we speculate it may reflect the effect of statins on autophagy or apoptosis which we previously observed in cells exposed to statins [6,7], and which are also cause by ivermectin. Alternatively, ivermectin has also been shown to reduce levels of mutated p53 [35]. Although mutations in p53 can inactivate its function as a tumour suppressor, some mutations endow a gain-of-function phenotype. We have previously shown that three gain-of-function variants of p53 can drive expression of HMGCR [7]. Thus, inhibition of mutant p53 by ivermectin may reduce the level of HMGCR and converge with direct inhibition of HMGCR by pitavastatin to bring about a synergistic reduction in HMGCR activity and inhibit the mevalonate pathway.

In these studies, we used ivermectin at a concentration at which it only had a small effect when used on its own. The rationale for this was two-fold. Firstly, if the effect of the combination was greater than that of pitavastatin alone, it would provide very clear evidence of synergy because, under these conditions, ivermectin alone had essentially no effect. Secondly, we wanted to maximize the chance of identifying a concentration that was clinically achievable in patients but also sufficient to be synergistic with pitavastatin. Although we observed synergy, the concentration of ivermectin was again higher than that achieved in patients treated with plasma. Thus, it is unlikely that directly transplanting the current treatment strategy for using ivermectin as an anti-parasitic agent to use as an anti-cancer agent in combination will succeed. Rather, it seems likely that methods will be necessary to allow the accumulation of ivermectin specifically in tumour tissues. Nanoparticular formulations are one potential route to achieve this [36]. The enhanced permeability and retention (EPR) effect allows nanoparticles to accumulate in tumours due to defects in the tumour vasculature and lack of clearance of the nanoparticles into the lymphatic system. Drugs contained in such nanoparticles, therefore, can accumulate in the tumour tissue. Excitingly, nanoparticle formulations of ivermectin have been developed [37] and these warrant evaluation in combination with pitavastatin.

These data suggest that in certain ovarian cancer patients, ivermectin may be usefully combined with pitavastatin to reduce the dose of pitavastatin required to treat patients with ovarian cancer. However, further work to identify the patients likely to respond and to improve ivermectin delivery to tumours will be necessary.

## Figures and Tables

**Figure 1 diseases-11-00049-f001:**
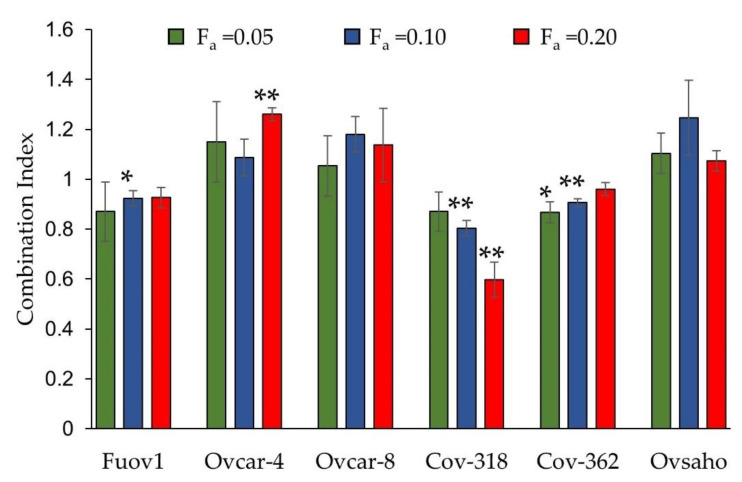
Evaluation of the combination of pitavastatin and ivermectin in cell growth assays. The indicated cells were simultaneously exposed for 72 h to a range of pitavastatin concentrations with a fixed concentration of ivermectin which, on its own, inhibited cell growth by 5%, 10%, and 20%. These concentrations were respectively: FUOV-1 (4.4 µM, 6.2 µM, and 8.8 µM); Ovcar-4 (4.2 µM, 5.9 µM, and 8.4 µM); Ovcar-8 (4.6 µM, 6.1 µM, and 8.3 µM); COV-318 (4.4 µM, 6.6 µM, and 9.0 µM); COV-362 (4.5 µM, 6.0 µM, and 8.0 µM); Ovsaho (4.6 µM, 6.1 µM, and 8.3 µM). Combination indices (mean ± S.D., n = 3–4) differed significantly from unity where indicated (* *p* ≤ 0.05; ** *p* < 0.01, paired *t*-test).

**Figure 2 diseases-11-00049-f002:**
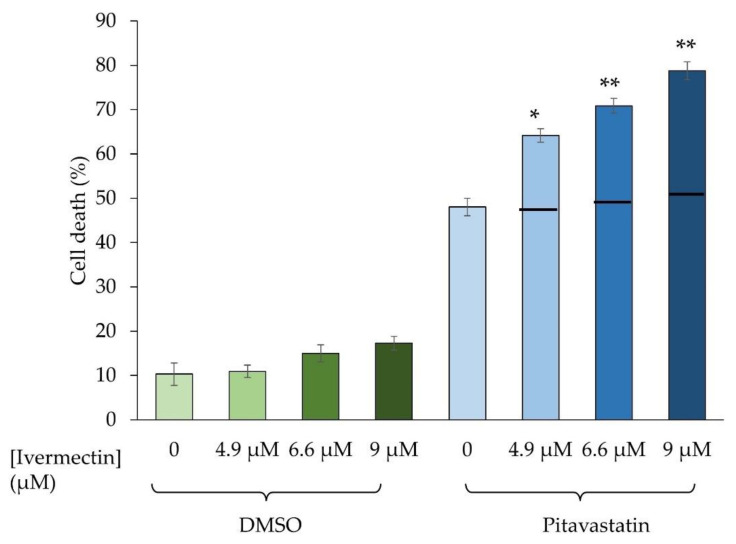
Evaluation of the combination of pitavastatin and ivermectin on cell viability. COV-318 cells were exposed for 72 h to ivermectin (4.9 µM, 6.6 µM, or 9.0 µM) in combination with 6.2 μM pitavastatin. The number of viable and dead cells was determined by staining with trypan blue and the percentage of dead cells calculated (mean ± S.D., n = 3). There were significantly (paired *t*-test, *, *p* < 0.01; **, *p* < 0.005) more dead cells in samples exposed to the combination than would be expected if the drugs acted additively (estimated using the Bliss independence criterion and shown as a horizontal black bar on the graph).

**Figure 3 diseases-11-00049-f003:**
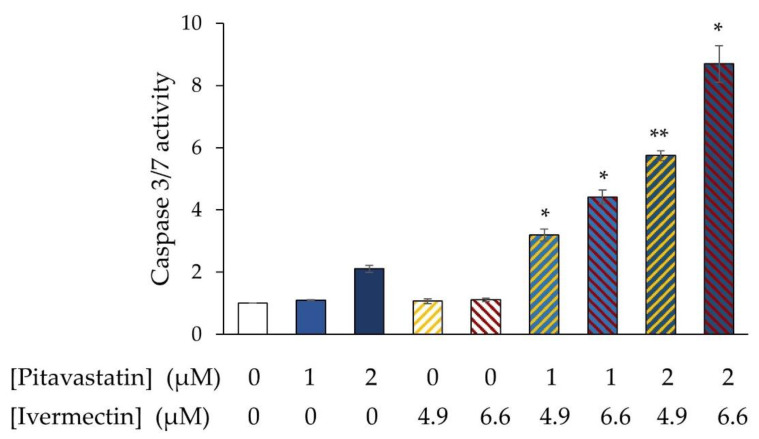
Evaluation of the combination of pitavastatin and ivermectin on caspase-3/7 activity. Cov-318 cells were exposed to pitavastatin (1 or 2 μM) with/without ivermectin (4.9 µM, 6.6 µM, or 9.0 µM) and after 72 h caspase 3/7 activity was measured. To control for potential effects of the drugs on cell number, the caspase-3/7 activity was normalised to the number of surviving cells estimated at the same time by staining a separate plate with SRB. The results are expressed as a proportion of the activity measured in untreated cells (mean ± SD, n = 3). The results were significantly different from those in cells treated with pitavastatin alone where indicated (*, *p* < 0.005, **, *p* < 0.001, paired *t*-test).

**Figure 4 diseases-11-00049-f004:**
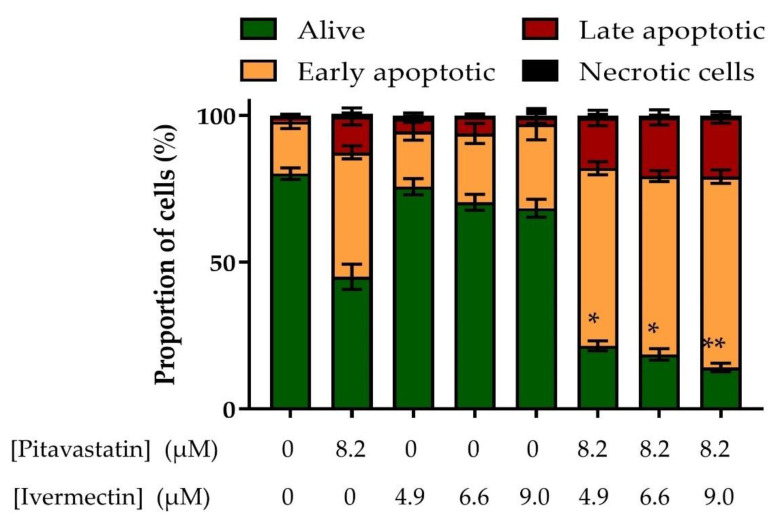
Evaluation of the combination of pitavastatin and ivermectin on annexin V/propidium iodide labelling. The effect of ivermectin combinations on annexin V/propidium iodide-staining. Cov-318 cells were treated with the indicated concentration of ivermectin, with/without pitavastatin (8.2 μM) for 48 h; the cells were labelled with annexin V and propidium iodide and analysed by flow cytometry. The results are expressed as a fraction of the total number of cells analysed (mean ± SD, n = 3). The results were significantly different from those in cells treated with pitavastatin alone where indicated (*, *p* < 0.05, **, *p* < 0.01, paired *t*-test).

**Table 1 diseases-11-00049-t001:** Potency of ivermectin in cell growth assays with ovarian cancer cells. The cells were exposed to a range of concentrations of pitavastatin for 72 h and the relative number of surviving cells determined by staining with SRB. The IC_50_s of ivermectin are reported (mean ± S.D., n = 3).

Cell Line	Ivermectin IC_50_ (μM)
Cov318	15 ± 2
Fuov-1	13 ± 2
Cov-362	15 ± 1
Ovcar-4	12 ± 1
Ovcar-8	16 ± 1
Ovsaho	15 ± 2

## Data Availability

Data is contained within the article.

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
