# Peer review of "Ivermectin Augments the Anti-Cancer Activity of Pitavastatin in Ovarian Cancer Cells"

_diseases, 2023, doi:10.3390/diseases11010049_

Round 1

Reviewer 1 Report

The work is average but may be improved by the inclusion of the following suggestions.

1.      English should be improved throughout the manuscript.

2.      Quantitative information should be provided in the abstract.

3.      The concussion should be concise and to the points indicating the application of the work.

4.      The novelty of the work should be established.

5.      Please write one paragraph in the introduction on cancer, in general, and you can consult the following articles to make this manuscript more useful to the readers.

Frontiers in Microbiology 12, 638609 (2021); Spectrochimica Acta Part A: Molecular and Biomolecular Spectroscopy 225, 117453 (2020); Future Med. Chem., 5: 135-146 (2013).; Future Med. Chem., 5, 961-978 (2013).; Med. Chem., 9, 11-21 (2013).;

6.  Please provide error graphs in the figure; where are required.

7.      Please improve the quality of the Figures.

8.      Please compare your results with previous studies and mention clearly how your work is important in comparison to already been reported.

Author Response

We would like to thank the referees for their helpful comments which we believe have helped improve the manuscript. In response we have made the changes noted below. Please note that lines numbers refer to the version of the manuscript with “track changes” activated. For your convenience, I have reproduced the questions raised by the referees and these are shown in bold.

  1. English should be improved throughout the manuscript.

We have changed “does” to “dose” in the abstract.

Line 53 – we have rewritten “function” as “functional”

Line 58 “associated” has been included

Line 59 deleted “but”

Line 124 A comma has been inserted after “here”.

line 152 We have corrected the spelling of “respectively”

Line 156 We have replaced “drug” with “drugs”

Line 178 We have removed a sentence fragment that should have previously been deleted

Line 203 We have moved the word “the” to its correct place in the sentence.

Line 223 we have inserted “that would be expected”

Line 295 – we have inserted the word “may”

We have inserted a comma after “thus” in line 293

We have deleted a superfluous “s” in “reflects” (line 318)

We have deleted a superfluous space on lines 29,106, 116, 135, 141, 180, 204

We have changed line 305 to read “In the cell growth assays reported here,”

Line 316 We have inserted “the”

There are number of instances throughout the manuscript where we have used as capital “I” instead of a lower case “I” for ivermectin.

We have used the English spelling for catalyses, programme tumour, analysed, haemocytometer and normalised rather than the American spellings.

Apart from that, I don’t believe there any other major English language errors – I am a native speaker and have written many manuscripts. If there is something I have overlooked, however, I would be happy to correct it.

  1. Quantitative information should be provided in the abstract.

We have quantified the increase in cell death and caspase activation in the abstract. We had already included quantitative information for the combination index and the IC50 of ivermectin

  1. The concussion should be concise and to the points indicating the application of the work.

The conclusion appears on lines 325-325. We have made some modifications to the text to make the application of our work clearer.

  1. The novelty of the work should be established.

We already commented on this in the discussion, but after reflecting on the referee’s comments, we think it would better to have a clearer statement of this in the introduction. We have inserted this sentence on line 115 “To our knowledge, the combination of pitavastatin and ivermectin alone has not previously been reported”

  1. Please write one paragraph in the introduction on cancer, in general, and you can consult the following articles to make this manuscript more useful to the readers.

Frontiers in Microbiology 12, 638609 (2021); Spectrochimica Acta Part A: Molecular and Biomolecular Spectroscopy 225, 117453 (2020); Future Med. Chem., 5: 135-146 (2013).; Future Med. Chem., 5, 961-978 (2013).; Med. Chem., 9, 11-21 (2013).;

We have written an introductory paragraph on cancer in general as the referee helpfully suggested.

  1. Please provide error graphs in the figure; where are required.

There are already error bars on all the graphs, so we aren’t sure what the reviewer is requesting we change here

  1. Please improve the quality of the Figures.

We have now included colour figures in the manuscript and we have increased the resolution of the figures.

  1. Please compare your results with previous studies and mention clearly how your work is important in comparison to already been reported.

We have included an additional sentences in the first paragraph of the discussion (lines 269-279) to compare our work to previous studies. In addition, a very substantial proportion of the discussion compares our results to the pharmacokinetics of ivermectin that have already been reported by others, and the impact that this has on the potential use of ivermectin.

Reviewer 2 Report

Dear Editor,

In the current manuscript, the authors experimentally demonstrated that Ivermectin enhances the anti-cancer activity of pitavastatin in ovarian cancer cells. The manuscript was well performed and well documented. It can be accepted after some corrections given below:

1.       Some informative information abou avermectins including ivermectin should be given in introduction section. For this aim, the authors can evaluate the following studies: “The impact of some avermectins on lactoperoxidase from bovine milk. International Journal of Food Properties, 19(6), 2016, 1207–1216”, “The effects of some avermectins on bovine carbonic anhydrase enzyme. Journal of Enzyme Inhibition and Medicinal Chemistry, 31(5), 2016, 773–778”.

2.       Line 152: The drawing quality of equation should be improved.

3.       Line 188: “5%, 10% or 20%” should be given as “5, 10 or 20%”.

4.       For integrity, it would be better if the font in the figures is the same as in the text.

5.       It would be better if the figures were shifted to the right in the text.

Author Response

We would like to thank the referees for their helpful comments which we believe have helped improve the manuscript. In response we have made the changes noted below. Please note that lines numbers refer to the version of the manuscript with “track changes” activated. For your convenience, I have reproduced the questions raised by the referees and these are shown in bold.

In the current manuscript, the authors experimentally demonstrated that Ivermectin enhances the anti-cancer activity of pitavastatin in ovarian cancer cells. The manuscript was well performed and well documented. It can be accepted after some corrections given below:

  1. Some informative information abou avermectins including ivermectin should be given in introduction section. For this aim, the authors can evaluate the following studies: “The impact of some avermectins on lactoperoxidase from bovine milk. International Journal of Food Properties, 19(6), 2016, 1207–1216”, “The effects of some avermectins on bovine carbonic anhydrase enzyme. Journal of Enzyme Inhibition and Medicinal Chemistry, 31(5), 2016, 773–778”.

We have included some additional information and cited these articles in the introduction, as the referee suggests on line 109-111

  1. Line 152: The drawing quality of equation should be improved.

We have rewritten this with Microsoft equation editor.

  1. Line 188: “5%, 10% or 20%” should be given as “5, 10 or 20%”.

We have corrected this as the referee suggests.

  1. For integrity, it would be better if the font in the figures is the same as in the text.

We have changed the font in the figures to match the font selected by the journal

  1. It would be better if the figures were shifted to the right in the text.

We agree, but that is part of the document preparation process the journal performs. We have done our best to centre it in the revised manuscript.

Round 2

Reviewer 1 Report

Revision is not complete; especially the novelty, language and necessary literature citation.

Either reject or give one more chance.

Author Response

I am discussing this matter with the Editor.